# Reduced Levels of Neurosteroids in Cerebrospinal Fluid of Amyotrophic Lateral Sclerosis Patients

**DOI:** 10.3390/biom14091076

**Published:** 2024-08-28

**Authors:** Chiara Lucchi, Cecilia Simonini, Cecilia Rustichelli, Rossella Avallone, Elisabetta Zucchi, Ilaria Martinelli, Giuseppe Biagini, Jessica Mandrioli

**Affiliations:** 1Department of Biomedical, Metabolic and Neural Sciences, University of Modena and Reggio Emilia, 41125 Modena, Italy; lucchi.chiara@unimore.it (C.L.); elisabetta.zucchi@unimore.it (E.Z.); martinelli.ilaria@aou.mo.it (I.M.); jessica.mandrioli@unimore.it (J.M.); 2Department of Neurosciences, Azienda Ospedaliero-Universitaria di Modena, 41126 Modena, Italy; ceciliasimonini24@gmail.com; 3Department of Life Sciences, University of Modena and Reggio Emilia, 41125 Modena, Italy; cecilia.rustichelli@unimore.it (C.R.); rossella.avallone@unimore.it (R.A.)

**Keywords:** neurosteroids, amyotrophic lateral sclerosis, cerebrospinal fluid

## Abstract

Produced by the mitochondria and endoplasmic reticulum, neurosteroids such as allopregnanolone are neuroprotective molecules that influence various neuronal functions and regulate neuroinflammation. They are reduced in neurodegenerative diseases, while in the Wobbler mouse model, allopregnanolone and its precursor progesterone showed protective effects on motor neuron degeneration. This single-center case-control study included 37 patients with amyotrophic lateral sclerosis (ALS) and 28 healthy controls. Cerebrospinal fluid (CSF) neurosteroid levels were quantified using liquid chromatography–electrospray tandem mass spectrometry and compared between the two cohorts. Neurosteroid concentrations have been correlated with neuroinflammation and neurodegeneration biomarkers detected through an automated immunoassay, along with disease features and progression. Pregnenolone, progesterone, allopregnanolone, pregnanolone, and testosterone levels were significantly lower in ALS patients’ CSF compared to healthy controls. A significant inverse correlation was found between neurofilament and neurosteroid levels. Neurosteroid concentrations did not correlate with disease progression, phenotype, genotype, or survival prediction. Our study suggests the independence of the disease features and its progression, from the dysregulation of neurosteroids in ALS patients’ CSF. This neurosteroid reduction may relate to disease pathogenesis or be a consequence of disease-related processes, warranting further research. The inverse correlation between neurosteroids and neurofilament levels may indicate a failure of compensatory neuroprotective mechanisms against neurodegeneration.

## 1. Introduction

Amyotrophic lateral sclerosis (ALS) is a devastating disease characterized by an unusually complex genetic and clinical landscape, implying that diverse pathomechanisms might be involved in its onset and progression. A key finding is mitochondrial dysfunction, which involves changes in mitochondrial shape, energy production, and calcium levels [1]. Mitochondria are essential for neuronal cell activities, including steroid synthesis, which is started in the inner mitochondrial membrane by the enzyme CYP11A1 through the conversion of cholesterol to pregnenolone, and then continued in the mitochondria-associated endoplasmic reticulum membrane [2]. Indeed, changes in mitochondrial membrane potential and energy production resulted in an impaired steroidogenesis, thus highlighting the need for mitochondrial integrity for this physiological function [3].

Steroids are powerful modulators of brain activities and, especially neurosteroids, are regarded as neuroprotective molecules that influence various biological functions in neuronal cells, such as neuronal growth, synapse formation, and neurogenesis [4]. For this reason, the most representative neurosteroid, namely allopregnanolone, is under investigation as a putative drug to treat neurodegenerative diseases such as fragile X-associated tremor/ataxia syndrome [5]. Interestingly, the enzyme known as type 10 17β-hydroxysteroid dehydrogenase (17β-HSD10), which inactivates allopregnanolone [6], is increased in the brain of subjects with Alzheimer’s disease and their respective animal models [7]. Another key feature of neurosteroids is their ability to regulate neuroinflammation, which is present in all neurodegenerative diseases, including ALS, by potentially suppressing microglial activation and reducing proinflammatory cytokine gene expression [8]. Microglia, but also macrophages and lymphocytes, express γ-aminobutyric acid type A (GABA_A_) receptors [9] that can be stimulated by allopregnanolone to reduce the release of pro-inflammatory mediators [10], thus providing a mechanism by which neurosteroids regulate neuroinflammation. Through GABA_A_ receptors, neurosteroids can inhibit nuclear factor kappa-light-chain-enhancer of activated B cells (NF-κB) activation, the production of inflammatory mediators, and monocyte chemoattractant protein-1 (MCP-1) secretion, all mechanisms that have been described in ALS [11,12].

Neurosteroid synthesis has been reported to be reduced in neurodegenerative diseases, including Alzheimer’s [13], Parkinson’s disease [14], and multiple sclerosis [15], whereas a recent study identified reduced dihydrotestosterone levels in ALS patients’ cerebrospinal fluid (CSF), hinting at decreased aromatase activity [16]. It is also interesting to observe that the reduction in brain levels of neurosteroids could be involved in the disease’s progression, as suggested by a preclinical study using the Wobbler mouse model. In this model, mice developed a recessive mutation in the gene coding for the Vps54 component of the Golgi apparatus, leading to the degeneration of their motor neurons (MNs) in a manner very similar to that observed in patients with ALS [17]. Notably, exogenously administered neurosteroids like progesterone [18] and allopregnanolone [19] were able to counteract the progression of MN degeneration, thus suggesting a possible role of these neurosteroids in the disease’s pathophysiology.

In view of this evidence, we investigated the changes in central levels of various neurosteroids, including pregnenolone sulfate, pregnenolone, progesterone, 5α-dihydroprogesterone, allopregnanolone, pregnanolone, and testosterone, by analyzing the CSF of patients with ALS compared to healthy subjects. Our aim was to disclose any possible reduction in neurosteroid levels to tentatively establish an involvement of these molecules in the disease’s progression.

## 2. Methods

In this single-center case-control study, we enrolled 37 patients with probable or definite ALS, admitted to the Neurology Unit of Modena University Hospital from 2015 to 2020. These patients underwent lumbar puncture (LP) as part of their diagnostic process and had at least 0.7 mL of CSF and serum stored at the Neurobiobank of Modena. Healthy controls (HCs; *n* = 28) were individuals admitted within the same period for suspected neurological diseases that were later unconfirmed. These individuals donated their CSF to the Neurobiobank of Modena.

The study received approval from the Ethical Committee of Area Vasta Emilia Nord (file number: 63/2022). All participants provided informed consent for the LP procedure, biobanking, and participation in research studies. We gathered a comprehensive set of demographic and clinical variables for patients including onset site, genotype, comorbidities, diagnostic latency, body mass index (BMI), forced vital capacity (FVC%), weight loss, and the ALS Functional Rating Scale—Revised (ALSFRS-R) score at sampling and last observation. The progression rate from onset to sampling and from sampling to last observation was calculated as previously reported [20].

The time to generalization was defined as the duration from the onset to the spread of clinical signs from the spinal or bulbar localization to both. Tracheostomy-free survival was calculated from the onset to tracheostomy or death. Patients were classified as “slow progressors” if their progression rate at sampling was ≤0.9 points/month, and as “fast progressors” if the progression rate was >0.9 points/month.

Samples were processed within two hours after collection and centrifuged for 10 min at 1300× *g*, and the supernatant was aliquoted and stored at −80 °C. Measurements of CSF and serum neurofilaments, SerpinA1, Triggering Receptor Expressed on Myeloid cells 2 (TREM-2), Chitinase 3-like 1 (CHI3L1), blood cell count, and C reactive protein were conducted as previously described [21].

Neurosteroid quantification in CSF was performed using liquid chromatography–electrospray tandem mass spectrometry (LC-MS/MS). All standards and chemicals were purchased from Sigma-Aldrich (St. Louis, MO, USA). Internal standards with isotope labeling were the following: 5α-pregnan-3α-ol-20-one-17α, 21, 21, 21-d4, and sodium pregnenolone-17α, 21, 21, 21-d4 sulfate. All solvents for high-performance LC/electrospray ionization tandem MS (HPLC/ESI-MS/MS) were LC-MS purity grade, whereas other solvents used for sample preparation were analytical grade). A stock solution was serially diluted with methanol to obtain calibration solutions at ten concentrations. The internal standard solution was prepared separately. Sample processing was performed as previously described [22]. Eluates were concentrated, derivatized with Amplifex Keto Reagent, and transferred in autosampler vials for LC-MS/MS analysis. The chromatographic separation was performed on an Agilent 1200 Series Binary Pump (Agilent, Waldbronn, Germany). MS detection was performed using an Agilent QQQ-MS/MS(6410B) triple quadrupole mass analyzer equipped with an ESI ion source (Agilent), operating in the positive mode [22].

Given the non-normal and left-skewed distribution of neurosteroid concentrations, the Mann–Whitney U test for two-group comparisons and the Kruskal–Wallis test for multiple-group comparisons were used, with a post hoc Dunn correction for multiple comparisons. Correlations between biomarkers and clinical variables were evaluated using Spearman’s rank correlation test. To specifically evaluate the role of age and sex on neurosteroid concentrations in ALS patients and healthy controls, we performed a robust regression analysis. Age and sex were included as independent variables, and neurosteroid concentrations were the dependent variable. An interaction term between age and sex was also examined to assess whether the effect of sex on neurosteroid levels varied with age. Tracheostomy-free survival was analyzed using univariate Cox regression analysis, followed by multivariate Cox proportional hazard modeling using a stepwise backward selection method with a retention criterion of *p* < 0.1.

Data analysis was conducted with STATA (Stata Corp (2017) Stata Statistical Software: Release 15. College Station, TX, USA: StataCorp LLC).

## 3. Results

### 3.1. Participants’ Characteristics

Thirty-seven ALS patients (17 women, mean age 52.08 years [SD 11.56]) with a mean time from disease onset of 10.71 months [SD 6.53]), and 28 HCs (18 women, mean age 49.16 years [SD 16.71]) were included. Among the ALS patients, eight patients had a bulbar onset. The mean ALSFRS-R score, progression rate, and FVC at sampling were 40.97 [SD 4.83], 1.57 points/month [SD 1.69], and 94.82% [SD 24.61], respectively. The mean diagnostic delay and time to generalization were 7.79 [SD 5.30] and 13.85 months [SD 13.10].

### 3.2. Biomarker Concentration in ALS Patients and HCs

Table 1 and Figure 1 show neurosteroid, neurofilament, and other biomarker concentrations in ALS patients and HCs. In the CSF of ALS patients, pregnenolone sulfate levels (0.022 [0.021–0.049] in women and 0.052 [0.049–0.064] in men) and testosterone levels (0.009 [0.008–0.10] in women and 0.014 [0.014–0.016] in men) differ significantly between sexes (*p* = 0.0034 and *p* = 0.0016, respectively).

After adjusting for sex and age at sampling, the difference between ALS patients and HCs remained significant for pregnenolone sulfate (*p* = 0.020), pregnenolone (*p* < 0.001), progesterone (*p* = 0.040), allopregnanolone (*p* = 0.042), pregnanolone (*p* = 0.004), and testosterone (*p* = 0.002), with lower values in ALS patients (Table 2).

### 3.3. Correlations between Neurosteroids, Measures of Disease Progression, and Biomarkers of Neurodegeneration and Neuroinflammation in ALS Patients

Within the ALS group, no correlation was found between neurosteroids and clinical and demographical features, except for a negative correlation between 5α-dihydroprogesterone and age at sampling (r = −0.30, *p* = 0.016), and also for pregnenolone sulfate and time to generalization (r = −0.39, *p* = 0.022). We found an inverse correlation between serum neurofilament light and pregnenolone (r = −0.38, *p* = 0.004), progesterone (r = −0.27, *p* = 0.04), pregnanolone (r = −0.43, *p* < 0.001), and testosterone (r = −0.41, *p* = 0.05). Similar correlations were present among the same neurosteroids and other CSF/serum neurofilaments. A further inverse correlation was found between testosterone and SerpinA1 in CSF (r = −0.80, *p* = 0.01) and serum TREM2 (r = −0.82, *p* = 0.02), and between pregnanolone and TREM2 (r = −0.51, *p* = 0.003) and CHI3L1 (r = −0.41, *p* = 0.01) in CSF (Table 3).

Neurosteroid levels were not modified among patients classified as slow or fast progressors (Table 4).

### 3.4. Neurosteroids, Biomarkers of Neurodegeneration and Neuroinflammation, Clinical Features, and ALS Survival

The factors influencing tracheostomy-free survival in ALS patients, as determined through univariate Cox regression analysis, are presented in Table 5. Besides clinical factors, the possible prognostic factors among biomarkers were represented by CSF and serum NfL (HR 1.00, 95% CI 100–1.00, *p*  =  0.001 and HR 1.01, 95% CI 100–1.00, *p*  <  0.001), CSF NfH (HR 1.00, 95% CI 100–1.00, *p*  =  0.030), allopregnanolone (HR 3.24; 95% CI 1.28–8.19; *p* = 0.013), lymphocytes (HR 0.28, 95% CI 0.11–0.96, *p* = 0.006), and neutrophil/lymphocyte ratio (HR 0.04, 95% CI 0.00–0.46, *p* = 0.010).

The multivariate analysis of survival showed that independent prognostic factors related to worse tracheostomy-free survival were neutrophil/lymphocyte ratio (HR 1.415, 95% CI 1.11–1.80, *p* = 0.005), serum NfL (HR 1.007, 95% CI 1.00–1.01, *p* = 0.003), and progression rate at sampling (HR 4.806, 95% CI 1.98–11.66, *p* = 0.001).

Neurosteroid levels did not influence tracheostomy-free survival according to multivariate Cox regression analyses (Table 5).

## 4. Discussion

In our pivotal study, overall decreased levels of neurosteroids starting from the precursors of steroidogenic synthesis were found in the CSF of ALS patients.

The reduction in most neurosteroids in the CSF of ALS patients, without association with phenotypic features or disease progression, may suggest that neurosteroid-dysregulated production at a central nervous system level is a widespread phenomenon in ALS. This phenomenon could be the consequence of reduced synthesis in the peripheral sources of neurosteroids, i.e., steroidal glands. This was not assessed in our investigation, but previous studies failed in finding a relationship between peripheral and central levels of neurosteroids, respectively, in patients with status epilepticus or multiple sclerosis [23,24].

Neurosteroids, like pregnenolone and progesterone, are pivotal for neuroprotection, enhancing cell survival, and diminishing neuroinflammation. The precursor pregnenolone activates Sig-1Rs across various brain cells to promote repair mechanisms and the microglia’s switch to the M2 neuroprotective phenotype [25]. Similarly, progesterone counters neurological damage and shifts microglia from a pro- to an anti-inflammatory state, while 5α-dihydroprogesterone and allopregnanolone reduce inflammation and support neuroprotection and myelin repair [26]. Additionally, allopregnanolone resulted in being neuroprotective through the induction of autophagy in addition to GABA_A_ receptor enhancement [27].

Although no clinical correlation was found with neurosteroids, an intriguing inverse relation with neurofilaments was observed. Being that neurofilaments established biomarkers of neurodegeneration in ALS [28], it might be speculated that the observed reduction in neurosteroids reflects the failure or exhaustion of a compensatory mechanism [29]. This could be explained by mitochondrial malfunctioning, since these organelles are indispensable for steroidogenesis [2,3] and, instead, have been found to be damaged in ALS [1]. However, neurosteroids are initially produced in mitochondria, but later their synthesis is continued at the interface between the mitochondria and endoplasmic reticulum [2]. In our study, we found a reduced availability of neurosteroids directly synthetized in the mitochondria, i.e., pregnenolone and progesterone, and also of those not produced in these organelles, such as allopregnanolone and pregnanolone [30]. Moreover, the lack of changes in 5α-dihydroprogesterone levels suggests that multiple defects in different steps of the steroid metabolic pathway could occur in ALS patients, so as to impair most but not all the steroidal production.

Furthermore, the inverse correlation between known microglial biomarkers such as SerpinA1, TREM2, CHI3L1, and pregnanolone and testosterone may raise important questions concerning the binary relation between microglial activation following stress and steroidogenic synthesis. We recently showed that human microglia significantly increase allopregnanolone production following oxidative damage [22], indicating a rapid microglial response to injury through specific neurosteroid production. Conversely, Balan and colleagues have shown that both microglia and macrophages, when treated with allopregnanolone post-lipopolysaccharide exposure, exhibit a reduced release of pro-inflammatory mediators [10], highlighting allopregnanolone production promotion as a potential compensatory mechanism for maintaining brain-microenvironment homeostasis.

Despite the low sample size, our results suggest that the concomitant increase in neuroinflammation biomarkers and decrease in neurosteroids in the CSF of ALS patients could be an initial attempt to balance the neuroinflammation by microglia that is running out of a toxic phase. Higher levels of SerpinA1 were found in ALS fast-progressor patients, underlining this imbalance in the regulation of neuroinflammation [21].

Another interesting consideration concerns sex-specific differences in testosterone and pregnenolone sulfate levels in ALS patients, a finding never replicated in healthy subjects and other conditions such as status epilepticus [23]. In an animal model of multiple sclerosis, it was previously demonstrated that precursor pregnenolone synthesis in the spinal cord was altered in a sex-specific way and according to the progression of the pathology [31]. Our observation requires validation in a larger cohort of ALS patients, but if confirmed, it could indicate a sex-specific dimorphic response in steroidogenesis inherent to the pathobiology of ALS.

Investigating ALS models further could clarify if our findings represent an early pathological mechanism or a secondary phenomenon following microglial activation [22], while a deep understanding of neurosteroid profiles in ALS patients may reveal endogenous neuroprotective mechanisms amenable to pharmacological intervention. Additionally, our findings could be relevant for ALS comorbidities, such as depression. Barone and colleagues demonstrated that peripheral inflammatory biomarkers are positively linked with the severity of psychiatric symptoms, whereas peripheral GABAergic neurosteroids show either no relation or an inverse relation [32]. Interestingly, treatment with allopregnanolone increased neurosteroid levels in postpartum depression patients, but these changes were not directly correlated with the improvement in depression scores, which was instead associated with a reduction in inflammatory cytokines [33]. This overall evidence suggests that neuroinflammation can be targeted by increasing neurosteroids to alleviate the multiple consequences of its effects.

## 5. Conclusions 

In conclusion, our study highlights a significant reduction in neurosteroid levels in the CSF of ALS patients, independent of disease progression or phenotypic features. This widespread reduction points to a potential dysfunction in neurosteroid synthesis within the central nervous system, possibly linked to mitochondrial dysfunction. Additionally, the observed inverse relationship between neurosteroids and neurofilaments supports the hypothesis of a compromised neuroprotective response in ALS.

Overall, our findings underscore the complexity of neurosteroid regulation in ALS and its potential role in disease pathogenesis. The intricate relationship between neurosteroids and neuroinflammation, particularly the involvement of microglia, remains a critical area for future research. A deeper understanding of these mechanisms could pave the way for therapeutic interventions aimed at enhancing endogenous neuroprotective processes, potentially mitigating the effects of neurodegeneration in ALS.

## Figures and Tables

**Figure 1 biomolecules-14-01076-f001:**
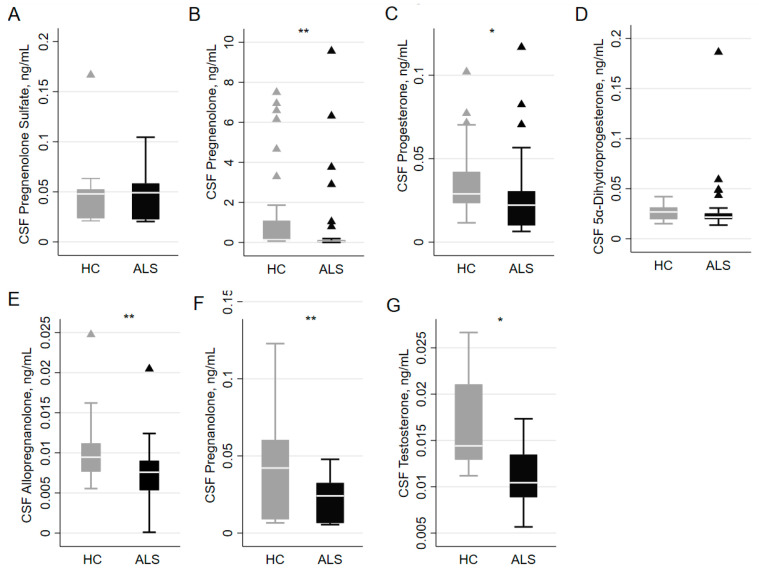
CSF concentrations (ng/mL) of different NSs in ALS and HC groups. In detail from left to right, (**panel A**) shows CSF concentration of pregnenolone sulfate, (**panel B**) of pregnenolone, (**panel C**) of progesterone, (**panel D**) of 5α-dihydroprogesterone, (**panel E**) of allopregnanolone, (**panel F**) of pregnanolone, and (**panel G**) of testosterone, respectively, in ALS patients (black) and HCs (gray). Mann–Whitney U test, * means *p* < 0.05, ** means *p* < 0.01.

**Table 1 biomolecules-14-01076-t001:** Comparison of neurosteroids and other biological variable concentrations between ALS patients and HCs.

Biological Variable, Fluid	N	ALS Patients, Median [IQR]	N	HCs, Median [IQR]	*p* Value
Pregnenolone sulfate, CSF (ng/mL)	37	0.0491 [0.0224–0.0532]	28	0.0478 [0.0235–0.0525]	0.9051
Pregnenolone, CSF (ng/mL)	37	0.0629 [0.0380–0.1023]	28	0.1283 [0.0931–1.088]	<0.0001
Progesterone, CSF (ng/mL)	37	0.0221 [0.0099–0.0305]	28	0.0289 [0.0233–0.0422]	0.0153
5α-dihydroprogesterone, CSF (ng/mL)	37	0.0215 [0.0196–0.0253]	28	0.0267 [0.0194–0.0314]	0.1310
Allopregnanolone, CSF (ng/mL)	33	0.0076 [0.0053–0.0090]	27	0.0094 [0.0076–0.0112]	0.0076
Pregnanolone, CSF (ng/mL)	36	0.0241 [0.0065–0.0325]	28	0.0421 [0.0089–0.0604]	0.0006
Testosterone, CSF (ng/mL)	17	0.0104 [0.0089–0.0135]	8	0.0144 [0.0129–0.0210]	0.0104
NfL, CSF (pg/mL)	37	6677 [3905–10,282]	20	434 [248–630]	<0.0001
NfL, serum (pg/mL)	37	115 [80–138]	20	9.20 [5.27–18.65]	<0.0001
NfH, CSF (pg/mL)	34	4012 [2471–7775]	25	549 [383–636]	<0.0001
NfH, serum (pg/mL)	33	1292 [639–2131]	17	147 [80–218]	<0.0001
CHI3L1, CSF (pg/mL)	26	72.53 [100.90–46.48]	10	86.38 [94.78–45.63]	0.9437
CHI3L1, serum (pg/mL)	26	32.89 [42.96–24.68]	10	38.98 [75.37–32.28]	0.0930
SerpinA1, CSF (μg/mL)	26	4.76 [2.01–7.83]	10	3.64 [2.75–4.77]	0.4280
SerpinA1, serum (μg/mL)	26	1048 [617–1464]	10	1344 [1008–3048]	0.0861
TREM2, CSF (ng/mL)	22	18.73 [14.28–24.34]	10	16.91 [11.85–20.44]	0.6256
TREM2, serum (ng/mL)	23	19.55 [15.76–28.05]	10	16.89 [11.22–27.61]	0.4807
Lymphocytes, blood (thousand/mmc)	33	1.89 [1.43–2.18]	22	2.07 [1.78–2.51]	0.1563
Neutrophils, blood (thousand/mmc)	37	3.58 [3.15–4.91]	25	3.74 [2.99–4.65]	0.8240
Neutrophil/lymphocyte ratio	33	2.06 [1.68–2.90]	22	1.93 [1.49–2.40]	0.2715
Monocytes, blood (thousand/mmc)	33	0.48 [0.43–0.61]	22	0.51 [0.42–0.63]	0.6426
CRP, blood (mg/dL)	33	0.5 [0.5–0.5]	17	0.5 [0.5–1]	0.0683

**Table 2 biomolecules-14-01076-t002:** Regression analysis examining the impact of sex and age on neurosteroid concentrations in ALS and HC groups.

Biological Variable, Fluid	Coefficient	95% CI	*p* Value
Pregnenolone sulfate, CSF ALS vs. HC Sex Age	−0.0123 −0.0044 −0.00027	−0.022 to −0.002 −0.016 to 0.007 −0.0005 to 0.000	0.020 0.459 0.049
Pregnenolone, CSF ALS vs. HC Sex Age	−0.0815 −0.0098 0.00014	−0.113 to −0.050 −0.046 to 0.027 −0.0007 to 0.0009	<0.0001 0.593 0.734
Progesterone, CSF ALS vs. HC Sex Age	−0.0096 −0.0074 0.00003	−0.019 to −0.005 −0.018 to 0.003 −0.0002 to 0.0002	0.040 0.168 0.800
5α-dihydroprogesterone, CSF ALS vs. HC Sex Age	−0.0043 0.00014 −0.00007	−0.087 to 0.001 −0.005 to 0.053 −0.0002 to 0.0000	0.059 0.953 0.229
Allopregnanolone, CSF ALS vs. HC Sex Age	−0.0020 0.00092 −0.00001	−0.004 to −0.0001 −0.001 to 0.003 −0.0001 to 0.0000	0.042 0.396 0.629
Pregnanolone, CSF ALS vs. HC Sex Age	−0.02177 −0.00097 −0.00029	−0.036 to −0.007 −0.017 to 0.016 −0.0007 to 0.0001	0.004 0.908 0.136
Testosterone, CSF ALS vs. HC Sex Age	−0.0053 −0.0012 0.00016	−0.008 to −0.002 −0.005 to 0.003 0.0000 to 0.0003	0.002 0.545 0.016

**Table 3 biomolecules-14-01076-t003:** Correlations between neurosteroids, clinical features, and biomarkers of neurodegeneration and neuroinflammation in ALS patients.

	Pregnenolone Sulfate, CSF (ng/mL)		Pregnenolone, CSF (ng/mL)		Progesterone, CSF (ng/mL)		5α-di-hydroprogesterone, CSF (ng/mL)		Allopregnanolone, CSF (ng/mL)		Pregnanolone, CSF (ng/mL)		Testosterone, CSF (ng/mL)	
	Spearman Rho Correlation	*p* Value	Spearman Rho Correlation	*p* Value	Spearman Rho Correlation	*p* Value	Spearman Rho Correlation	*p* Value	Spearman Rho Correlation	*p* Value	Spearman Rho Correlation	*p* Value	Spearman Rho Correlation	*p* Value
Age at sampling	−0.23	0.07	−0.12	0.32	−0.03	0.82	−0.30	0.02	−0.16	0.24	−0.20	0.11	0.01	0.95
ALSFRS-R total score at sampling	−0.16	0.34	.004	0.98	0.03	0.87	−0.23	0.18	−0.21	0.25	0.22	0.19	−0.23	0.38
Progression rate at sampling *	0.19	0.26	0.09	0.61	−0.13	0.43	0.22	0.18	0.25	0.16	0.03	0.86	0.25	0.34
Time to generalization ^§^	−0.39	0.022	−0.29	0.10	0.23	0.19	0.03	0.85	−0.25	0.17	−0.22	0.21	−0.27	0.34
ALSFRS-R total score at last observation	−0.13	0.45	−0.04	0.83	−0.15	0.36	−0.17	0.32	−0.03	0.88	−0.19	0.27	0.007	0.98
Progression rate at last observation ^#^	0.26	0.12	−0.04	0.81	0.03	0.84	0.001	1.00	0.32	0.07	−0.15	0.38	0.07	0.80
NfL, CSF (pg/mL)	0.05	0.70	−0.38	0.004	−0.23	0.08	−0.17	0.21	−0.16	0.27	−0.35	0.008	−0.39	0.06
NfL, serum (pg/mL)	0.05	0.72	−0.38	0.004	−0.27	0.04	−0.23	0.08	−0.14	0.31	−0.43	<0.001	−0.41	0.047
NfH, CSF (pg/mL)	−0.06	0.64	−0.38	0.003	−0.20	0.13	−0.24	0.07	−0.21	0.12	−0.51	<0.001	−0.57	0.004
NfH, serum (pg/mL)	0.13	0.33	−0.42	0.001	−0.34	0.009	−0.15	0.27	−0.14	0.32	−0.24	0.07	−0.53	0.010
CHI3L1, CSF (pg/mL)	−0.22	0.20	−0.02	0.92	0.09	0.61	−0.25	0.13	0.10	0.58	−0.41	0.01	−0.08	0.83
CHI3L1, serum (pg/mL)	−0.21	0.23	−0.00	1.00	0.06	0.73	−0.05	0.77	−0.04	0.82	0.08	0.67	−0.15	0.70
SerpinA1, CSF (μg/mL)	−0.08	0.64	−0.13	0.47	−0.16	0.36	−0.24	0.15	−0.29	0.09	−0.21	0.24	−0.80	0.010
SerpinA1, serum (μg/mL)	0.19	0.28	0.25	0.15	0.33	0.05	0.20	0.25	0.34	0.05	0.15	0.39	−0.20	0.61
TREM2, CSF (ng/mL)	−0.25	0.17	0.02	0.90	−0.01	0.95	−0.32	0.07	−0.08	0.67	−0.51	0.003	−0.80	0.10
TREM2, serum (ng/mL)	−0.05	0.78	−0.05	0.77	0.22	0.21	0.06	0.73	−0.05	0.80	−0.02	0.91	−0.82	0.023
Lymphocytes, serum (thousand/mL)	−0.02	0.90	−0.07	0.62	−0.05	0.71	0.13	0.34	−0.09	0.54	0.17	0.22	−0.38	0.08
Neutrophils, serum (thousand/mL)	−0.03	0.84	−0.11	0.42	0.004	0.97	−0.02	0.88	−0.06	0.66	−0.01	0.94	−0.26	0.22
Neutrophil/lymphocyte ratio	−0.12	0.38	−0.08	0.58	−0.03	0.83	0.26	0.06	−0.09	0.52	−0.24	0.08	−0.06	0.80
Monocytes, serum (thousand/mL)	0.07	0.58	0.11	0.41	0.19	0.17	0.08	0.55	−0.10	0.48	0.03	0.73	0.14	0.52
CRP, serum (mg/dL)	−0.16	0.26	0.12	0.41	0.04	0.78	−0.08	0.59	−0.28	0.05	0.03	0.81	0.33	0.14

* The progression rate at sampling is determined by the monthly decline in the ALSFRS-R total score. This is calculated by starting with a total score of 48 at the onset of symptoms, then subtracting the total score at sampling, and dividing this difference by the number of months from symptom onset to sampling. ^§^ Time to generalization is defined as the period from symptom onset to when clinical signs spread from either spinal or bulbar localization to both areas. ^#^ The progression rate at sampling is determined by the monthly decline in the ALSFRS-R total score from sampling to the last observation. This is calculated by dividing the difference in the total score between these two points by the number of months from sampling to the last observation.

**Table 4 biomolecules-14-01076-t004:** Comparison of neurosteroid levels between slow and fast progressors among ALS patients.

Biological Variable, Fluid	N	Slow Progressors, Median [IQR]	N	Fast Progressors, Median [IQR]	*p* Value
Pregnenolone sulfate, CSF (ng/mL)	22	0.0482 [0.0221–0.0530]	15	0.0499 [0.0245–0.0616]	0.171
Pregnenolone, CSF (ng/mL)	22	0.0606 [0.0329–0.102]	15	0.0662 [0.0380–0.103]	0.656
Progesterone, CSF (ng/mL)	22	0.0232 [0.0169–0.0319]	15	0.0151 [0.00875–0.0299]	0.426
5α-di-hydroprogesterone, CSF (ng/mL)	22	0.0215 [0.0165–0.0238]	15	0.0219 [0.0206–0.0257]	0.484
Allopregnanolone, CSF (ng/mL)	19	0.00671 [0.00524–0.00880]	12	0.00882 [0.00644–0.00998]	0.148
Pregnanolone, CSF (ng/mL)	21	0.0216 [0.00648–0.0312]	15	0.0282 [0.00662–0.0329]	0.610
Testosterone, CSF (ng/mL)	11	0.00923 [0.00821–0.0135]	6	0.0114 [0.0104–0.0141]	0.171

**Table 5 biomolecules-14-01076-t005:** Univariate and multivariate Cox regression analysis of tracheostomy-free survival in ALS patients of the study.

Variable	Univariate Analysis HR 95% CI *p* Value			Multivariate Analysis HR 95% CI *p* Value		
Sex, male	1.42	0.56–3.57	0.45			
Age at onset, y	1.03	0.99–1.07	0.13			
Diagnostic delay, m	0.90	0.82–0.99	0.040			
Onset Lower limb Upper limb Bulbar	1 0.54 1.42	reference 0.18–1.58 0.50–4.06	0.259 0.507			
Presence of FTD	1.20	0.40–3.56	0.75			
Weight loss at diagnosis, Kg	1.13	1.05–1.23	0.001			
*C9ORF72* expansion	0.75	0.17–3.26	0.70			
ALSFRS-R score at sampling points	0.92	0.85–0.99	0.024			
Progression rate at sampling	2.75	1.51–5.02	0.001	4.81	1.98–11.66	0.001
FVC at diagnosis, %	0.97	0.95–0.98	0.001			
Time to generalization, m	0.94	0.90–0.99	0.027			
Pregnenolone sulfate, CSF *	1.66	0.69–3.98	0.26			
Pregnenolone, CSF *	1.34	0.56–3.21	0.507			
Progesterone, CSF *	1.24	0.52–2.99	0.62			
5α-dihydroprogesterone, CSF *	0.60	0.25–1.43	0.25			
Allopregnanolone, CSF *	3.24	1.28–8.19	0.013			
Pregnanolone, CSF *	1.95	0.77–4.91	0.16			
Testosterone, CSF *	1.45	0.24–8.80	0.68			
NfL, CSF (pg/mL)	1.00	1.00–1.00	0.001			
NfL, serum (pg/mL)	1.01	1.00–1.01	<0.001	1.01	1.00–1.01	0.003
NfH, CSF (pg/mL)	1.00	1.00–1.00	0.030			
NfH, serum (pg/mL)	1.00	1.00–1.00	0.58			
CHI3L1, CSF (pg/mL)	1.00	0.99–1.01	0.65			
CHI3L1, serum (pg/mL)	1.00	0.99–1.01	0.47			
SerpinA1, CSF (μg/mL)	1.12	0.98–1.29	0.079			
SerpinA1, serum (μg/mL)	1.00	1.00–1.00	0.59			
TREM2, CSF (ng/mL)	1.03	0.97–1.11	0.31			
TREM2, serum (ng/mL)	0.98	0.91–1.05	0.56			
Lymphocytes, blood (thousand/mmc)	0.28	0.11–0.96	0.006			
Neutrophils, blood (thousand/mmc)	1.14	0.88–1.48	0.32			
Neutrophil/Lymphocyte ratio	0.04	0.00–0.46	0.010	1.41	1.11–1.80	0.005
Monocytes, blood (thousand/mmc)	0.95	0.03–28.07	0.975			
CRP, blood (mg/dL)	1.31	0.39–4.41	0.66			

* In the univariate survival analysis, neurosteroids were evaluated as independent factors using the median CSF values as cutoff points rather than their continuous values. The cutoff values were as follows: pregnenolone sulfate: 0.0491 ng/mL; pregnenolone: 0.0629 ng/mL; progesterone: 0.0221 ng/mL; 5α-dihydroprogesterone: 0.0215 ng/mL; allopregnanolone: 0.0076 ng/mL; pregnanolone: 0.0241 ng/mL; and testosterone: 0.0104 ng/mL.

## Data Availability

The data that support the findings of this study are available from the corresponding author (giuseppe.biagini@unimore.it) to external researchers who provide methodologically sound scientific proposals and whose proposed use of the data has been approved by an independent review committee identified for this purpose. Responses to requests will be given in two months. A materials transfer and/or data access agreement with the study promoter will be required for accessing shared data.

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
