# Peer review of "Reduced Levels of Neurosteroids in Cerebrospinal Fluid of Amyotrophic Lateral Sclerosis Patients"

_biomolecules, 2024, doi:10.3390/biom14091076_

Round 1

Reviewer 1 Report

Comments and Suggestions for Authors

To authors

1)        In the introduction section, the authors discuss the Wobble mouse model. Could the authors briefly describe the phenotype? The readers are not informed as to what this animal model is used for.

2)        Section 3.2: The authors describe the results stratified by sex and age. However, the table does not include these results. Please add these results as a new table in an appendix.

3)        Table 1: The authors do not describe all the results of NSs concentrations.

4)        Table 1: Plasma values ​​are a reflection of those obtained in CSF. In the future, could the authors consider plasma NSs measurements as reliable diagnostic tools?

5)        Table 3.3: “...except for a negative correlation between 5α-dihydroprogesterone and age at sampling (r=-0.30, p=0.016),...”.

This is not indicated in Table 2. Furthermore, in the whole section 3.3. you do not describe all the results. Please, the authors should improve the description of all the results in Table 2.

6)        Section 3.3.: "...Unlike neutrophils/lymphocytes ratio (HR 1.415, 95%CI 1.11-1.80,  p=0.005), serum NfL (HR 1.007, 95%CI 1.00-1.01, p=0.003) and rate progression at sampling (HR 4.806, 95%CI 1.98-11.66, p=0.001), neurosteroid levels did not influence trache-139 ostomy-free survival at multivariate Cox regression analyses....".

Where are these values ​​located? Authors are also invited to indicate the values ​​in the table or include them in an appendix.

7)        Discussion: The authors comment that the decrease in NSs values ​​are not associated with phenotypic characteristics or disease progression. This is relative because the authors are demonstrating increased motor neuron death from the significant production of the motor neuron biomarker in Tables 1 and 2: neurofilaments (L and the phosphorylated form of H).

In this case, have the authors considered, or verified, what relationship the decrease in NSs has with the levels (in number or functional state) of mitochondria?

8)        In the introduction, a more detailed description of NSs and their relationship with neurodegenerative diseases should be provided.

9)        Although sex hormones are produced by both sexes (in greater or lesser quantities depending on the gender), the authors should better specify in the results whether the reduction in the concentration of SNs is due to the sample size resulting from gender differences or, in fact, due to the pathology.

10)   Are the authors aware that in presymptomatic states of the disease (with animal models of ALS) the NSs are higher? If so, it could be a compensatory mechanism to slow the onset of the disease (rather than increasing survival).

Author Response

REVIEWER #1

1) In the introduction section, the authors discuss the Wobble mouse model. Could the authors briefly describe the phenotype? The readers are not informed as to what this animal model is used for.

Answer: We added few sentences to illustrate the mouse model (lines 73-78): “In this model, mice developed a recessive mutation in the gene coding for the Vps54 component of Golgi apparatus, leading to degeneration of motor neurons (MNs) in a manner very similar to that observed in patients with ALS [17]. Notably, exogenously administered neurosteroids like progesterone [18] and allopregnanolone [19] were able to counteract the progression of MN degeneration, thus suggesting a possible role of these neurosteroids in the disease pathophysiology.”

2) Section 3.2: The authors describe the results stratified by sex and age. However, the table does not include these results. Please add these results as a new table in an appendix.

Answer: A new table (Table 2) is now available to illustrate the impact of sex and age on neurosteroid concentrations in ALS and HC.

3) Table 1: The authors do not describe all the results of NSs concentrations.

Answer: All concentrations of neurosteroids considered in our study (including pregnenolone sulfate, pregnenolone, progesterone, 5α-dihydroprogesterone, allopregnanolone, pregnanolone, and testosterone) were reported in Table 1. For the sake of clarity, we added few sentences at the end of introduction to list the neurosteroids evaluated in our study. Additionally, we reordered the analyzed neurosteroids according to their metabolic pathway in tables and figure. This order has been made consistent in all tables and figures.

4) Table 1: Plasma values ​​are a reflection of those obtained in CSF. In the future, could the authors consider plasma NSs measurements as reliable diagnostic tools?

Answer: Thank you for this suggestion. Indeed, peripheral steroids are metabolized to neurosteroids in the brain or directly enter in the pool of neurosteroids available to modulate the activity of neurons and glia. Thus, also for this reason steroidal glands are important modulators of brain functioning. However, we previously failed in showing a relationship between serum and cerebrospinal fluid levels in patients with status epilepticus (Meletti et al. 2017). Other independent studies also reported the same problem in finding a relationship between peripheral and central levels of neurosteroids (Caruso et al. 2014). This comment has been added to discussion (lines 246-250): “This phenomenon could be the consequence of reduced synthesis in the peripheral sources of neurosteroids, i.e. steroidal glands. This was not assessed in our investigation, but previous studies failed in finding a relationship between peripheral and central levels of neurosteroids, respectively in patients with status epilepticus or multiple sclerosis [23,24].”

5) Table 3.3: “...except for a negative correlation between 5α-dihydroprogesterone and age at sampling (r=-0.30, p=0.016),...”.  This is not indicated in Table 2. Furthermore, in the whole section 3.3. you do not describe all the results. Please, the authors should improve the description of all the results in Table 2.

Answer: we improved the above mentioned table, now named table 3, adding a row reporting the correlations among neurosteroids and age at sampling.

Furthermore we reported the results related to neurosteroids concentrations in slow and fast progresin g patients in a new table (table 4).

6)        Section 3.3.: "...Unlike neutrophils/lymphocytes ratio (HR 1.415, 95%CI 1.11-1.80,  p=0.005), serum NfL (HR 1.007, 95%CI 1.00-1.01, p=0.003) and rate progression at sampling (HR 4.806, 95%CI 1.98-11.66, p=0.001), neurosteroid levels did not influence tracheostomy-free survival at multivariate Cox regression analyses....".

Where are these values ​​located? Authors are also invited to indicate the values ​​in the table or include them in an appendix.

Answer: A new table (Table 5) is now available to illustrate univariate and multivariate Cox regression analysis of trachesotomy-free survival

7)        Discussion: The authors comment that the decrease in NSs values ​​are not associated with phenotypic characteristics or disease progression. This is relative because the authors are demonstrating increased motor neuron death from the significant production of the motor neuron biomarker in Tables 1 and 2: neurofilaments (L and the phosphorylated form of H).

In this case, have the authors considered, or verified, what relationship the decrease in NSs has with the levels (in number or functional state) of mitochondria?

Answer: Unfortunately, we had not the possibility to evaluate the changes in mitochondria in our patients, but we are aware of the importance and complexity of this hypothesis. Neurosteroids are initially produced in mitochondria, but later their synthesis is continued at the interface between the mitochondria and endoplasmic reticulum. In our study we found a reduced availability of neurosteroids directly synthetized in the mitochondria, i.e. pregnenolone, and of those not produced in mitochondria, such as allopregnanolone and pregnanolone. On the other hand, the lack of changes in 5α-dihydroprogesterone levels suggests that multiple defects in different steps of the metabolic pathway could occur in ALS patients. This comment has been added to discussion (lines 259-272).

8)        In the introduction, a more detailed description of NSs and their relationship with neurodegenerative diseases should be provided.

 Answer: We provided more details on the involvement of neurosteroids in neurodegeneration (line 52-57: “the most representative neurosteroid, namely allopregnanolone, is under investigation as a putative drug to treat neurodegenerative diseases such as fragile X-associated tremor/ataxia syndrome [5]. Interestingly, the enzyme known as type 10 17β-hydroxysteroid dehydrogenase (17β-HSD10), which inactivates allopregnanolone [6], is increased in the brain of subjects with Alzheimer’s disease and the respective animal models [7]”). We also restructured the introduction to meet the reviewer’s requirements.

9) Although sex hormones are produced by both sexes (in greater or lesser quantities depending on the gender), the authors should better specify in the results whether the reduction in the concentration of SNs is due to the sample size resulting from gender differences or, in fact, due to the pathology.

Answer: This is an important point which was addressed by stratifying data by sex, as illustrated in the newly added Table 2.

10)   Are the authors aware that in presymptomatic states of the disease (with animal models of ALS) the NSs are higher? If so, it could be a compensatory mechanism to slow the onset of the disease (rather than increasing survival).

Answer: Thank you for this question, indeed it will be interesting to investigate this point in the Wobbler model of ALS. We certainly will consider this important development of our investigation in future experiments.

Reviewer 2 Report

Comments and Suggestions for Authors

This manuscript reported a single-center case-control study indicating a reduction of the neurosteroids levels in cerebrospinal fluid (CSF) of amyotrophic lateral sclerosis (ALS) patients. The reduction of neurosteroids shows a correlation with the increase of neurodegeneration markers (i.e. Neurofilament) and some neuroinflammation markers (Serpjn A1, TREM2, CHI3L1), but not with disease progression or clinical symptoms or stages. This study provides solid clinical evidence supporting the association between the reduction of neurosteroids with neurodegeneration in ALS, which will motivate the investigation of the role of neurosteroids in ALS pathogenesis as well as its therapeutic potential. The data were well analyzed and presented. The writing is succinct and clear. Suggest for publication.

Author Response

REVIEWER #2

This manuscript reported a single-center case-control study indicating a reduction of the neurosteroids levels in cerebrospinal fluid (CSF) of amyotrophic lateral sclerosis (ALS) patients. The reduction of neurosteroids shows a correlation with the increase of neurodegeneration markers (i.e. Neurofilament) and some neuroinflammation markers (Serpjn A1, TREM2, CHI3L1), but not with disease progression or clinical symptoms or stages. This study provides solid clinical evidence supporting the association between the reduction of neurosteroids with neurodegeneration in ALS, which will motivate the investigation of the role of neurosteroids in ALS pathogenesis as well as its therapeutic potential. The data were well analyzed and presented. The writing is succinct and clear. Suggest for publication.

Answer: Thank you very much for the appreciation of our work.

Reviewer 3 Report

Comments and Suggestions for Authors

The manuscript “Reduced neurosteroid levels in cerebrospinal fluid of amyotrophic lateral sclerosis patients” by Lucchi et al. effectively highlights significant differences in the levels of neurosteroids, such as progesterone, pregnenolone, pregnanolone, allopregnanolone, and testosterone, between ALS patients and healthy controls. The robustness of these findings is underscored by the fact that the differences remained significant even after adjusting for sex and age. The inverse correlations observed between neurofilament levels and several neurosteroids suggest a potential link between these biomarkers and neurodegenerative processes. However, neurosteroid levels did not appear to influence tracheostomy-free survival, indicating that while the neurosteroids can differentiate between ALS patients and controls, they may not have a direct impact on survival outcomes. Additionally, the lack of strong correlations between neurosteroids and clinical progression metrics highlights the need for further research with larger participant cohorts to better understand their roles in ALS progression and prognosis. Please address the following issues:

Table 1 and Figure 1F: Correct the term ‘solfate’ to ‘sulfate’.

Table 2: Reformat Table 2 as it is currently difficult to follow.

Line 118: Clarify how sex and age were included in the analysis. What specific statistical method was used for adjustment? Was sex treated as a categorical variable and age as a continuous variable, or was a different approach applied?

Discussion: Barone et al. (Psychoneuroendocrinology. 2023;157:106359. doi:10.1016/j.psyneuen.2023.106359) demonstrated that peripheral inflammatory biomarkers are positively linked with psychiatric symptom severity, while peripheral GABAergic neuroactive steroids show either no relation or an inverse relation. Additionally, while allopregnanolone infusion altered neurosteroid levels in postpartum depression patients, these changes were not directly correlated with improved depression scores. Instead, improvements were associated with reduced inflammatory cytokines due to allopregnanolone's inhibition of toll-like receptor activation (Patterson et al., Neuropsychopharmacology. 2024;49(1):67-72. doi:10.1038/s41386-023-01721-1). Could a similar mechanism be at play in ALS patients? Please consider adding this to your discussion.

Author Response

REVIEWER #3

The manuscript “Reduced neurosteroid levels in cerebrospinal fluid of amyotrophic lateral sclerosis patients” by Lucchi et al. effectively highlights significant differences in the levels of neurosteroids, such as progesterone, pregnenolone, pregnanolone, allopregnanolone, and testosterone, between ALS patients and healthy controls. The robustness of these findings is underscored by the fact that the differences remained significant even after adjusting for sex and age. The inverse correlations observed between neurofilament levels and several neurosteroids suggest a potential link between these biomarkers and neurodegenerative processes. However, neurosteroid levels did not appear to influence tracheostomy-free survival, indicating that while the neurosteroids can differentiate between ALS patients and controls, they may not have a direct impact on survival outcomes. Additionally, the lack of strong correlations between neurosteroids and clinical progression metrics highlights the need for further research with larger participant cohorts to better understand their roles in ALS progression and prognosis. Please address the following issues:

Table 1 and Figure 1F: Correct the term ‘solfate’ to ‘sulfate’.

Answer: Thank you for suggesting the correction of these inaccuracies. It has been done.

Table 2: Reformat Table 2 as it is currently difficult to follow.

Answer: Table 2 (now table 3) has been reorganized as a horizontal one. We hope that this facilitates the reading.

Line 118: Clarify how sex and age were included in the analysis. What specific statistical method was used for adjustment? Was sex treated as a categorical variable and age as a continuous variable, or was a different approach applied?

Answer: regression analyses was performed including sex as a categorical variable and age as a continuous variable (expressed by years). We added the results in Table 2, and specified the analyses in the methods section (lines 130-135)

Discussion: Barone et al. (Psychoneuroendocrinology. 2023;157:106359. doi:10.1016/j.psyneuen.2023.106359) demonstrated that peripheral inflammatory biomarkers are positively linked with psychiatric symptom severity, while peripheral GABAergic neuroactive steroids show either no relation or an inverse relation. Additionally, while allopregnanolone infusion altered neurosteroid levels in postpartum depression patients, these changes were not directly correlated with improved depression scores. Instead, improvements were associated with reduced inflammatory cytokines due to allopregnanolone's inhibition of toll-like receptor activation (Patterson et al., Neuropsychopharmacology. 2024;49(1):67-72. doi:10.1038/s41386-023-01721-1). Could a similar mechanism be at play in ALS patients? Please consider adding this to your discussion.

Answer: Thank you for the suggestion. These findings are now included in discussion (lines 296-308).

Round 2

Reviewer 1 Report

Comments and Suggestions for Authors

I appreciate the authors for their efforts to respond in detail and make the required improvements.